# Shiga Toxin 2 Triggers C3a-Dependent Glomerular and Tubular Injury through Mitochondrial Dysfunction in Hemolytic Uremic Syndrome

**DOI:** 10.3390/cells11111755

**Published:** 2022-05-26

**Authors:** Simona Buelli, Monica Locatelli, Claudia Elisa Carminati, Daniela Corna, Domenico Cerullo, Barbara Imberti, Luca Perico, Maurizio Brigotti, Mauro Abbate, Carlamaria Zoja, Ariela Benigni, Giuseppe Remuzzi, Marina Morigi

**Affiliations:** 1Istituto di Ricerche Farmacologiche Mario Negri IRCCS, Centro Anna Maria Astori, Science and Technology Park Kilometro Rosso, Via Stezzano 87, 24126 Bergamo, Italy; monica.locatelli@marionegri.it (M.L.); ce.carminati87@gmail.com (C.E.C.); daniela.corna@marionegri.it (D.C.); domenico.cerullo@marionegri.it (D.C.); barbara.imberti@marionegri.it (B.I.); luca.perico@marionegri.it (L.P.); mauro.abbate@marionegri.it (M.A.); carlamaria.zoja@marionegri.it (C.Z.); ariela.benigni@marionegri.it (A.B.); giuseppe.remuzzi@marionegri.it (G.R.); marina.morigi@marionegri.it (M.M.); 2Department of Experimental, Diagnostic and Specialty Medicine, University of Bologna, 40126 Bologna, Italy; maurizio.brigotti@unibo.it

**Keywords:** podocytes, proximal tubular epithelial cells, complement, C3a/C3aR signaling, mitochondrial damage

## Abstract

Shiga toxin (Stx)-producing *Escherichia coli* is the predominant offending agent of post-diarrheal hemolytic uremic syndrome (HUS), a rare disorder of microvascular thrombosis and acute kidney injury possibly leading to long-term renal sequelae. We previously showed that C3a has a critical role in the development of glomerular damage in experimental HUS. Based on the evidence that activation of C3a/C3a receptor (C3aR) signaling induces mitochondrial dysregulation and cell injury, here we investigated whether C3a caused podocyte and tubular injury through induction of mitochondrial dysfunction in a mouse model of HUS. Mice coinjected with Stx2/LPS exhibited glomerular podocyte and tubular C3 deposits and C3aR overexpression associated with cell damage, which were limited by C3aR antagonist treatment. C3a promoted renal injury by affecting mitochondrial wellness as demonstrated by data showing that C3aR blockade reduced mitochondrial ultrastructural abnormalities and preserved mitochondrial mass and energy production. In cultured podocytes and tubular cells, C3a caused altered mitochondrial fragmentation and distribution, and reduced anti-oxidant SOD2 activity. Stx2 potentiated the responsiveness of renal cells to the detrimental effects of C3a through increased C3aR protein expression. These results indicate that C3aR may represent a novel target in Stx-associated HUS for the preservation of renal cell integrity through the maintenance of mitochondrial function.

## 1. Introduction

Shiga toxin (Stx)-producing *E. coli* (STEC) is a causative agent of Stx-associated hemolytic uremic syndrome (HUS), a disorder characterized by thrombocytopenia, microangiopathic hemolytic anemia, and acute renal failure, which predominantly affects infants and small children [1,2,3,4]. The outcome of Stx-associated HUS has improved over the last few decades, and the acute mortality rate in children is 1–4%. About 60–70% of patients recover completely from the acute phase but the remainder have varying degrees of sequelae [5]. Long-term renal sequelae have been reported in up to 20–40% of patients who experience proteinuria, glomerular injury and chronic kidney disease [5,6].

The kidney is one of the primary target organs severely affected by Stx as a result of the high levels of the host receptor globotriaosyl ceramide (Gb3) [7]. Upon binding on the cell surface and internalization, the toxin activates a cascade of intracellular signals that lead to glomerular endothelial inflammatory and thrombotic injury [8,9,10] as well as to podocyte and tubular cell dysfunction [11,12]. A large body of studies over the last three decades has shown that the hyperactivation of the complement system may contribute to the pathophysiology of Stx-associated HUS [13]. First clinical reports described that reduced serum levels of C3 were associated with the activation of the alternative pathway C3-convertase and with a consequent increase in C3b, C3c and C3d breakdown products [14,15,16]. Similar observations were made in a Swedish cohort of 10 children with active Stx-associated HUS exhibiting elevated plasma levels of C3a and soluble C5b-9 in the early phase of the disease [17], and the presence of C3 on platelet-leukocyte complexes and microparticles [18]. At the renal level, C3 deposition in the glomeruli of HUS pediatric patients was associated with thickening of the glomerular capillary wall, endothelial dysfunction and effacement of podocyte foot processes [19,20,21], further indicating that complement activation plays a prominent role in the pathogenesis of the glomerular lesions observed in the disease. In a mouse model of HUS obtained by Stx2/LPS-coinjection, we found that Stx2 was able to activate complement through the alternative pathway, favoring exuberant glomerular C3b deposits and the generation of C3a [22,23]. The latter was responsible for the loss of thromboresistance in glomerular endothelial cells [22] and the dysfunction of podocytes [23], both cell populations being highly sensitive to Stx cytotoxicity [8,24]. In experimental HUS, treatment with a C3a receptor (C3aR) antagonist counteracted glomerular endothelial and podocyte injury and restored renal function, thus identifying C3a as a key player in the development of kidney injury in HUS [22,23]. 

The main recognized cellular responses following the binding of C3a to the C3aR—a cell surface transmembrane G protein-coupled receptor—include the engagement of a wide range of signal transductors such as ERK1/2, PI3K/AKT and cAMP [25,26], leading to cytoskeletal changes [27] and increased expression of proinflammatory cytokines [28]. A recent report has highlighted the novel finding of a detrimental link between the activated C3a/C3aR axis and changes in mitochondrial homeostasis [29]. Specifically, in response to oxidative stress, human epithelial cells exhibited the transfer of C3aR from the cell plasma membrane to mitochondria, where the binding to C3a promoted a pathological mitochondrial Ca^2+^ influx and a reduction in mitochondrial respiration and ATP synthesis [29]. In the present study we sought to investigate whether the activation of the complement cascade, with the renal deposition of C3 and generation of C3a in response to Stx, could contribute to glomerular and tubular cell injury through the induction of mitochondrial dysfunction both in an experimental murine model of HUS and in cultured cells.

## 2. Materials and Methods

### 2.1. Experimental Model of Shiga Toxin-Associated HUS

Male C57BL/6 mice were obtained from Charles River Laboratories Italia (Lecco, Italy) and were housed in a specific pathogen-free facility with constant temperature on a 12:12-h light-dark cycle with free access to standard diet and water. Stx2 was purified as described previously [30,31]. The amount of LPS in the Stx2 preparation was very low (6.5 ng/mg of protein) as assessed by the Limulus assay. Animals were randomly allocated to the following groups: group 1 (*n* = 7), C57BL/6 mice given purified Stx2 (120 ng per mouse) plus LPS (60 μg per mouse; L2630, Merck, Darmstadt, Germany) by intraperitoneal injection to induce HUS; group 2 (*n* = 7), Stx2/LPS-treated mice given the C3a receptor antagonist SB290157 (Cayman Chemical, Ann Arbor, MI, USA) at a dose of 2.5 mg/kg [32], by intraperitoneal injection, 1 h before Stx2/LPS injection and twice a day for two days. Four mice injected with saline (group 3) served as controls. Mice were euthanized 48 h after injection of Stx2/LPS through CO_2_ inhalation and their kidneys were collected and processed for analysis. Before sacrifice, blood samples were collected for blood count and serum BUN measurements. In preliminary experiments the doses of Stx2 and LPS were established on the basis of their combined effects on platelet count (Stx2/LPS, 230 ± 48 vs. control, 823 ± 26 platelets 10^4^/μL, *p* < 0.01) and renal function assessed by BUN (Stx2/LPS, 31.8 ± 2.6 vs. control, 18.3 ± 1.0 mg/dL, *p* < 0.01) to obtain the same experimental condition previously described [22,23]. The dose of 60 μg LPS alone did not affect renal function (LPS, 19.4 ± 1.6 vs. control, 18.1 ± 0.9 mg/dL) but decreased platelet count (LPS, 425 ± 33 vs. control, 866 ± 89 platelets 10^4^/μL, *p* < 0.01), as previously described [22], consistent with the presence of the LPS receptor TLR4 on platelet surface.

### 2.2. Immunohistochemical Analyses in Renal Tissue 

For immunofluorescence studies, 3 µm-thick renal cryosections were fixed with cold acetone and incubated with 1% bovine serum albumin (BSA, Sigma-Aldrich, St. Louis, MO, USA) to block non-specific sites. To evaluate C3 deposits, the sections were incubated with FITC-conjugated goat anti-mouse C3 antibody (1:200; 55500, Cappel, Durham, NC, USA). For C3aR expression, double staining was performed with rabbit anti-C3aR antibody (1:100; LS-C382362, Lifespan BioSciences, Seattle, WA, USA) and Cy3-conjugated secondary antibody (1:100; 111-165-003, Jackson Immunoresearch Laboratories, West Grove, PA, USA), followed by rat anti-nestin (1:300; ab81462, Abcam, Cambridge, UK) and the appropriate FITC-conjugated secondary antibody (Jackson Immunoresearch Laboratories). The intraglomerular C3 and C3aR staining was quantified in 15–20 fields, whereas tubular staining was measured in 5–10 fields randomly acquired per sample using the ImageJ/Fiji ver 2.3 software (ImageJ, imagej.net/software/fiji/), and data were expressed as the percentage of positive staining on glomerular area or total area, respectively. 

Megalin, tubulin and cleaved caspase-3 expression were evaluated in 4% paraformaldehyde-fixed renal samples (3 µm thick) blocked in BSA 1% and then incubated with goat anti-megalin (1:50; sc16478, Santa Cruz Biotechnology, Dallas, TX, USA), mouse anti-tubulin (1:100; T9026, Sigma-Aldrich) or rabbit anti-cleaved caspase-3 (1:50; 9664, Cell Signaling Inc., Danvers, MA, USA) followed by the appropriate secondary antibody. For tubulin staining, antigen retrieval with citrate buffer was performed before BSA incubation. Megalin expression was evaluated in 5 fields per sample given a score between 0 and 4 (0, no alterations; 1, ≤25% of the tubules affected; 2, 26–50% of the tubules affected; 3, 51–75% of the tubules affected; 4, >75% of the tubules affected; the pattern of megalin expression was considered altered when thinner or discontinuous compared to that of control tubules) and results were expressed as an index, calculated as follow: [(n0 × 0) + (n1 × 1) + (n2 × 2) + (n3 × 3) + (n4 × 4)]/total proximal tubules counted, where n is the number of tubules in the field with a given score. Cleaved caspase-3 expression was quantified in 10 randomly acquired fields per sample using the Fiji software. Nuclei and cell membranes were counterstained with DAPI and FITC- or Cy5-labeled WGA-lectin, respectively. Negative controls were obtained by omitting the primary antibodies on adjacent sections. Samples were examined under a confocal inverted laser microscope (Leica TCS SP8, Leica Microsystems, Wetzlar, Germany). 

For immunoperoxidase analyses, formalin-fixed, paraffin-embedded kidney sections (3 µm thick) were subjected to antigen retrieval in a decloaking chamber with Rodent Decloacker buffer (RD913, Biocare Medical, Pacheco, CA, USA) and then incubated with Peroxidazed 1 (PX968, Biocare Medical) to quench endogenous peroxidases. After blocking of non-specific sites with Rodent Block M (RBM961, Biocare Medical), sections were incubated with the following primary antibodies: mouse anti-VDAC (1:250 and 1:400 for glomerular and tubular staining, respectively; ab186321, Abcam), rabbit anti-ATP5I (1:100 and 1:300 for glomerular and tubular staining, respectively; HPA035010, Sigma-Aldrich). The samples were then incubated with the appropriate HRP-Polymer (Biocare Medical). The staining was visualized by the addition of the Betazoid DAB Chromogen Kit solutions (BDB2004, Biocare Medical). Slides were finally counterstained with hematoxylin and observed through light microscopy (ApoTome, Axio Imager Z2, Zeiss, Oberkochen, Germany). Negative controls were obtained by omitting the primary antibody on adjacent sections. Glomerular and tubular VDAC and ATP5I expression was quantified in 15–20 (glomerular) or 10–15 (tubular) randomly acquired fields per sample using the Fiji software, and data were expressed as the percentage of positive staining on glomerular area or total area, respectively.

### 2.3. Histological and Ultrastructural Analyses 

Renal sections (3 µm thick) fixed in Duboscq-Brasil and embedded in paraffin were stained with periodic acid-Schiff (PAS) reagent (Bio-Optica, Milan, Italy) and assessed by light microscopy. Tubular lesions, evaluated as tubular dilation and cytoplasmic vacuolation in tubule segments and collecting ducts, were examined in 15–20 fields for each animal and were expressed with a score from 0 to 3 (mild, moderate and severe). For renal ultrastructural analysis, glutaraldehyde-fixed fragments of cortical kidney tissue were processed as previously described [33] and examined with a Philips Morgagni 268D transmission electron microscope (TEM; Philips, Brno, Czech Republic).

### 2.4. Estimation of Mitochondrial Size Measurements in Podocytes

Mitochondria were assessed with TEM and mitochondrial size measurements were obtained using the Fiji software. The major axes of the ellipse equivalent to the mitochondrion were determined, whereas the mitochondrial area was measured as the two-dimensional area in μm^2^. The morphology of at least 400–500 mitochondria was determined in podocytes of 6–7 glomeruli for 2–3 mice/group.

### 2.5. Morphometric Analysis of Mitochondria in Proximal Tubular Cells

Glutaraldehyde-fixed fragments of cortical kidney tissue were processed as previously described [33], and examined with TEM. Numerical density of mitochondria (N_V_, n/µm^3^) was estimated using morphometrical analysis according to Weibel [34], using an orthogonal grid digitally superimposed to 30 digitized electron microscope pictures of proximal tubules at ×7100 for each sample. Briefly, the mitochondrial profile area density (N_A_) was estimated by the ratio between the number of mitochondria and the proximal tubular area in the image calculated on the basis of grid points. Mitochondrial volume density (V_V_) was determined by the ratio of grid points falling over mitochondria divided by the total number of points of the grid container in proximal tubule section (*n* = 3 mice/group). N_V_ was then estimated for each animal using the formula [34]:N_V_ = (1/*β*) (N_A_^3/2^/V_V_^1/2^)
where *β* is the shape coefficient for ellipsoidal mitochondria, calculated from the ratio of the harmonic mean of major and minor axis of the mitochondrial sections measured on digital images. The mean mitochondrial volume was calculated for each animal (*n* = 3/group) as the ratio of mitochondrial volume density V_V_ and numerical density N_V_.

### 2.6. Cell Cultures and Incubation

Conditionally immortalized human podocytes were kindly provided by Dr P. Mathieson and Dr M.A. Saleem (Children’s Renal Unit and Academic Renal Unit, University of Bristol, Southmead Hospital, Bristol, UK) and cultured as previously described [23]. Briefly, cells were cultured under growth-permissive conditions at 33 °C in RPMI 1640 medium (21875034, Thermo Fisher Scientific, Waltham, MA, USA) supplemented with 10% fetal bovine serum (FBS; 10270106, Thermo Fisher Scientific), 1% ITS (insulin, transferrin, and sodium selenite) (41400045, Thermo Fisher Scientific) and 1% Pen-Strep (penicillin 100 U/mL plus 0.1 mg/mL streptomycin) (15140122, Thermo Fisher Scientific). To induce differentiation, podocytes were grown on rat tail-derived collagen type I and maintained in non-permissive conditions at 37 °C for at least 12 days. 

Human primary proximal tubule epithelial cells (RPTECs) were purchased from Lonza (CC-2553) and grown in REBM medium (CC-3191, Lonza, Basel, Switzerland) enriched with REGM SingleQuots^TM^ (CC-4127, Lonza) and 5% FCS. For all experiments, RPTECs were seeded 30,000 cells/cm^2^ and cells between the fifth and seventh passage were used.

For the experiments, podocytes and RPTECs were incubated with test medium (RPMI 1640 for podocytes, REBM medium for RPTECs, supplemented with 1% FBS) alone or in the presence of C3a 1 µM (A118, Complement Technology, Tyler, TX, USA) for 6 h. In selected experiments, cells were exposed to Stx2 (50 pM) alone for 15 h or for 24 h with the addition of C3a in the last 6 h of incubation.

Cell viability was evaluated in cells exposed to Stx2 at the concentration of 50 pM for 24 h. This dose did not affect the viability of podocytes (control, 21.8 ± 0.7 vs. Stx2, 22.1 ± 1.8 cells/field) as previously described [35], and of RPTECs (control, 53.7 ± 2.2 vs. Stx2, 47.7 ± 0.3 cells/field) thus indicating that 50 pM Stx2 is a subtoxic concentration in our setting.

### 2.7. Immunofluorescence Analysis in Cultured Cells

At the end of the incubations, podocytes and RPTECs were fixed in 2% paraformaldehyde and 4% sucrose, then permeabilized with 0.3% Triton X-100 (Sigma-Aldrich). Nonspecific binding sites were blocked with 2% FBS, 2% BSA and 0.2% bovine gelatin. Cells were incubated with a rabbit anti-C3aR (1:50; LS-C382362, Lifespan BioSciences, Seattle, DC, USA) antibody followed by a Cy3-conjugated secondary antibody (1:80; 711-165-152, Jackson ImmunoResearch Laboratories). Nuclei were counterstained with DAPI (Sigma-Aldrich). Negative controls, obtained by incubating cells with rabbit IgG (sc-3888, Santa Cruz) before Cy3-conjugated secondary antibody, exhibited no unspecific signal. Samples were examined using confocal microscopy (Leica TCS SP8, Leica Microsystems). The quantification of C3aR expression was performed on 10–15 random fields per sample. Specifically, the areas corresponding to the staining were measured in pixel^2^ using the ImageJ software and normalized for the number of nuclei identified by DAPI staining.

### 2.8. Mitochondrial Morphology in Cultured Cells

To evaluate mitochondrial morphology, podocytes and RPTECs were incubated with the fluorescent probe MitoTracker™ Red CMXRos (250 nM; M7512, Thermo Fisher Scientific) in the last 30 min of the stimuli. Nuclei were counterstained with NucBlue™ Live ReadyProbes™ Reagent (R37605, Thermo Fisher Scientific). At the end of the incubations, living cells were examined with confocal microscopy (Leica TCS SP8, Leica Microsystems). The percentage of podocytes and RPTECs with an altered mitochondrial pattern, in terms of fragmentation and/or perinuclear redistribution, on total cells per field was assessed in 10 randomly acquired fields per sample.

### 2.9. Western Blot Analysis

Equal amounts of 30 μg proteins from total extracts obtained by podocytes and RPTECs were separated by 12–15% SDS-PAGE and transferred to nitrocellulose membranes. After blocking with 5% BSA in tris-buffered saline (TBS) supplemented with 0.05% Tween-20 (Sigma-Aldrich), membranes were incubated with mouse anti-tubulin (1:2000; T9026, Sigma-Aldrich) or rabbit anti-SOD2 acetylated at lysine 68 (SOD2^AcK68^, 1:1000; ab137037, Abcam) and, on the same membrane, sheep anti-total SOD2 (1:1000; 574596, Calbiochem). The signals were visualized on an Odyssey^®^ FC Imaging System (LiCor, Lincoln, NE, USA) by infrared (IR) fluorescence using secondary goat anti-rabbit IRDye 800CW antibody (FE30926211, LiCor), goat anti-mouse IRDye 680LT (FE3680200, LiCor) or donkey anti-sheep HRP antibody (1:20,000; A3415, Sigma-Aldrich), as appropriate. Bands were quantified by densitometry using the Image Studio Lite ver 5.0 software (LI-COR Biotechnology, Lincoln, NE, USA). SOD2 acetylation was expressed as the ratio between SOD2^AcK68^ and total SOD2. Tubulin expression was normalized by Ponceau S staining.

### 2.10. Statistics

Results were expressed as mean ± SEM. Data analysis was performed with GraphPad Prism ver 9.3 Software (GraphPad Software Inc., San Diego, CA, USA). Comparisons were made using the two-sided unpaired Student’s *t* test or two-sided ANOVA with Tukey’s multiple comparisons post-hoc test, as appropriate. The statistical significance was defined as *p* < 0.05.

## 3. Results

### 3.1. Activation of Complement System and C3a/C3aR Axis Occurs in the Glomerular and Tubular Compartments in Stx2/LPS-Coinjected Mice 

To evaluate whether the complement system and the C3a/C3aR axis were activated in the kidneys of mice with experimental HUS, we studied the deposition of C3—which implies local complement activation and the consequent generation of C3a by the C3 cleavage [36]—and the expression of C3aR in the renal glomerular and tubular compartments of C57BL/6 mice coinjected with Stx2 and lipopolysaccharide (LPS). Forty-eight hours after injection, C3 markedly accumulated in the glomerular tuft showing a more widespread distribution compared to control mice given saline that exhibited C3 positivity only along the Bowman’s capsule (Figure 1A, upper panel). This abnormal deposition of complement in the glomerulus was also paralleled by a significant increase in linear C3 deposits along the tubular basement membrane, compared to control mice (Figure 1A, bottom panel). Mice injected with LPS (60 μg/mice) alone exhibited no increased C3 deposition at the glomerular or tubular level compared to controls (Appendix A). Weak staining for C3aR was observed in both glomerular and tubular structures of control mice (Figure 1B), whereas 48 h after Stx2/LPS injection, the expression of C3aR markedly increased in the glomerular tuft (Figure 1B, upper panel), mainly in nestin-stained podocytes (Appendix A), and in the tubules (Figure 1B, bottom panel).

### 3.2. C3aR Antagonist Counteracts Podocyte Injury by Reducing Mitochondrial Alterations in Stx2/LPS-Coinjected Mice

The pathogenic role of the C3a/C3aR axis at the glomerular level has been investigated by studying, in Stx2/LPS mice, the effects of the treatment with the C3aR antagonist SB290157. The inhibition of C3aR markedly limited renal function impairment in terms of reduced blood urea nitrogen (BUN) levels in C3aR antagonist-treated with respect to vehicle-treated mice (Appendix A). C3aR blockade also had a protective effect on podocyte loss as shown by the preservation of glomerular nestin expression in Stx2/LPS mice (Stx2/LPS + C3aR antagonist: 18.9 ± 0.6; Stx2/LPS + vehicle: 16.3 ± 0.9, *p* < 0.05; control 22.1 ± 0.4% glomerular area, *p* < 0.01).

In search of mechanisms underlying complement-mediated podocyte injury, we assessed whether Stx2/LPS-injected mice exhibited mitochondrial alterations in podocytes. Ultrastructural analysis showed the presence of short-rod shape mitochondria, unlike the elongated organelles present in the podocytes of control mice (Figure 2A, upper panel). Dysmorphic mitochondria often exhibited matrix swelling associated with disarrangement of the cristae (Figure 2A, bottom panel). Changes in mitochondrial morphology were then quantified and the morphometric analysis showed a reduction in both mitochondrial length and surface area in the podocytes of Stx2/LPS mice compared to controls (Figure 2B). In particular, the population of the smallest mitochondria (<0.2 μm) became more abundant following Stx2/LPS injection, whereas the longest mitochondria (>0.4 μm) were less represented compared to what was observed in control animals (Figure 2C). These results were paralleled by a decrease in mitochondrial mass (Figure 2D, upper panels), which was documented through the quantification of the mitochondrial marker Voltage-Dependent Anion Channel (VDAC), the most abundant protein of the outer membrane [37]. Alterations in mitochondrial structure and mass were accompanied by reduced glomerular levels of the mitochondrial ATP5I (Figure 2D, bottom panels), a subunit of the ATP synthase, which is known to play a key role in the formation and stabilization of mitochondrial cristae and energy production [38].

Treatment with C3aR antagonist markedly limited mitochondrial morphological changes in podocytes of Stx2/LPS mice (Figure 2A–C). Specifically, mitochondria exhibited an improvement of the matrix content (Figure 2A) associated with a normalization of their length and the area (Figure 2B,C). Moreover, a restoration of the glomerular mitochondrial mass and function occurred, as revealed by the significant increase in VDAC levels (Figure 2D, upper panels) and ATP5I expression (Figure 2D, bottom panels). These findings demonstrate that C3a is an important determinant in the development of mitochondrial dysfunction, which is one of the leading causes of podocyte injury in mice with experimental HUS.

### 3.3. C3aR Blockade Decreases Tubular Cell Injury by Limiting Mitochondrial Dysfunction in Stx2/LPS Mice

Studies have shown that in patients with Stx-associated HUS, renal tubular injury is present as the consequence of glomerular damage or due to a direct effect of Stx on the tubules [39]. In kidney cortex of Stx2/LPS mice stained with PAS we observed the presence of focal areas with moderate tubule dilatation and cytoplasm vacuolation (Appendix A). Apoptosis, assessed by increased staining for cleaved caspase 3, was evident in the tubules of Stx2/LPS mice (Figure 3A). Furthermore, we found the presence of thinner and discontinuous staining of megalin—a multiligand functional receptor of proximal tubular cells responsible for filtered protein reabsorption [40]—in the brush border of tubular cells compared to control mice (Figure 3B) indicating an impairment of proximal tubular function in Stx2/LPS mice. Treatment with C3aR antagonist resulted in a reduction in tubular cell apoptosis (Figure 3A) and a normalization of megalin expression at the luminal surface of the proximal tubules (Figure 3B).

At the ultrastructural level, remarkable fragmentation and swelling of mitochondria, which exhibited intense matrix loss and irregular arrangement of fragmented cristae, were found in proximal tubular epithelial cells of Stx2/LPS mice (Figure 4A, middle panel), compared to control animals showing elongated mitochondria arranged along the infoldings of the basal plasma membrane (Figure 4A, left panel). Furthermore, mitochondria of various sizes were observed in Stx2/LPS mice to be scattered in the cell cytoplasm without alignment and orientation (Figure 4A, middle panel). Morphometric data showed an increase in the mean mitochondrial volume associated with a reduction in the mitochondrial density over controls (Figure 4B). Notably, in Stx2/LPS mice, the C3aR blockade significantly limited mitochondrial abnormalities, restoring the mitochondrial structure and distribution, and reestablishing the normal spatial interactions with the basal plasma membrane of proximal tubules (Figure 4A, right panel). The structural retention of the tubular mitochondrial network in response to C3aR blockade was closely associated with the restoration of the mitochondrial VDAC and ATP5I protein expression (Figure 4C).

Since the intracellular mitochondrial distribution is under the control of a complex interconnected microtubule network [41], we investigated whether tubulin cytoskeletal disarrangement occurred in the tubular compartment of Stx2/LPS mice, possibly triggered by C3a. By immunofluorescence analysis, proximal tubular cells of control mice showed tubulin filaments organized in stripe-like structures running from the basal to the apical aspect of the tubular cells (Figure 5). In contrast, Stx2/LPS mice exhibited a disorganized tubulin pattern throughout the cytoplasm of most renal proximal tubular epithelial cells (Figure 5). Importantly, treatment with C3aR antagonist almost completely abrogated the disruptive effects that Stx2/LPS had on tubulin distribution (Figure 5).

Altogether, these findings establish a causal link between the activation of C3a/C3aR signaling and tubular damage via mitochondrial dysregulation.

### 3.4. Stx2 Enhances Podocyte and Tubular Cell Susceptibility to the Toxic Effect of C3a on Mitochondria In Vitro

Having established that Stx2/LPS mice overexpressed C3aR in both glomerular and tubular structures, we performed in vitro experiments to assess whether Stx2 affected C3aR expression in podocytes and tubular cells. Cultured human podocytes and renal proximal tubular epithelial cells (RPTECs) were exposed to control medium or Stx2 (50 pM). Immunofluorescence analysis demonstrated that Stx2 stimulation induced a significant increase in C3aR expression in both podocytes and RPTECs compared to control cells (Figure 6A).

A direct toxic effect of C3a on mitochondrial integrity was demonstrated in both cultured podocytes and RPTECs by MitoTracker assay. We found that the cytoplasmatic filamentous network of elongated mitochondria observed in control podocytes and RPTECs was altered upon exposure to C3a, with the occurrence of mitochondrial fragmentation and redistribution of fragmented mitochondria to the perinuclear region of both cell types (Figure 6B, middle panels). Notably, pretreatment of podocytes with Stx2 drastically potentiated the detrimental effects of C3a on mitochondrial injury (Figure 6B, right panels). Similarly, Stx2 pre-exposure greatly increased C3a-mediated mitochondrial damage and perinuclear redistribution in RPTECs compared to C3a alone (Figure 6B), suggesting that Stx2 increased the glomerular and tubular responsiveness to C3a. In this setting, some mitochondria exhibited a swollen morphology similar to that observed in mice injected with Stx2/LPS (Figure 6B, arrowhead). In parallel, we found that tubulin expression was reduced in podocytes and RPTECs following C3a stimulation (Figure 7A), indicating that changes in the tubulin network may account for the mitochondrial redistribution toward the perinuclear region induced by C3a.

Damaged mitochondria are the main source of intracellular ROS due to the decrease in their antioxidant activity [42]. The effect of C3a on the activity of the mitochondrial antioxidant enzyme superoxide dismutase 2 (SOD2), which plays a key role in the mitochondrial defense against ROS production [43] was evaluated by western blot analysis. We found that C3a stimulation increased the expression of the acetylated SOD2, the inactive form of the enzyme, in both podocytes and RPTECs (Figure 7B). Moreover, co-incubation of C3a with Stx2 further raised acetylated SOD2 levels (Figure 7B). These data indicate that Stx2 potentiates the effects of C3a decreasing the mitochondrial antioxidant defense of podocytes and RPTECs.

## 4. Discussion

In the present study, we described a novel mechanism through which Stx causes renal injury in experimental HUS. We found in Stx2/LPS mice that:(a)increased intraglomerular and tubular staining for C3 and C3aR overexpression were accompanied by podocyte loss and tubular cell damage;(b)mitochondrial abnormalities, in terms of altered morphology and distribution, as well as the reduction in mitochondrial VDAC and ATP5I expression, occurred in podocytes and proximal tubular cells;(c)the inhibition of the C3a/C3aR axis by treatment with a C3aR antagonist reduced podocyte and tubular cell phenotypic changes having normalized the ultrastructural and functional mitochondrial abnormalities.

In vitro, we showed that C3a directly affected the mitochondrial structural and functional activity and promoted oxidative stress in cultured podocytes and tubular cells. Notably, Stx2 pre-exposure increased the cell sensitivity to the detrimental effects of C3a, possibly by enhancing the C3aR protein expression.

A growing body of evidence has documented the dysregulation or overactivation of the complement system in HUS with increased plasmatic levels and intrarenal deposits of C3, C3a and C5b-9 in patients with active Stx-associated HUS [16,17]. Clinical reports, ranging from case series to cohort studies [44,45,46], have provided different results that seem to indicate that treatment with eculizumab, a humanized monoclonal antibody targeting complement C5, particularly in patients with severe neurological dysfunction, had beneficial effects if given early [47]. However, no robust evidence of the efficacy of C5 blockade in the management of the disease could be provided or disputed based on the current data [48,49]. In the present study, increased renal C3 and C3aR staining in the experimental model of HUS induced in mice by coinjection of Stx2/LPS, together with the evidence of the renoprotection of blocking the C3a/C3aR axis, highlight a novel unidentified mechanism, through which activation of the complement system induces renal damage. In Stx2/LPS mice, abnormal activation and overexpression of complement proteins in the glomerular compartment were found to be associated with a reduction in glomerular endothelial thromboresistance [22] and podocyte injury, the latter due to alterations of cytoskeletal components and proteins regulating cell–cell and cell–matrix interactions [23]. Our experimental studies were consistent with data from patients with Stx-associated HUS showing podocyte foot process impairment associated with glomerular C3 deposition [19,20], and support the hypothesis that the complement system plays a role in HUS and, in particular, that the activation of C3 and the generation of C3a also have a major effect on glomerular damage in HUS. A close relationship between the development of glomerular lesions and tubular injury, which may be predictive of the long-term renal outcomes, has been described in patients [39,50,51,52] and in preclinical models of HUS [39,52,53,54]. In line with several studies that show tubular dilatation and apoptosis in models of HUS [54,55,56,57], here we found focal areas with moderate tubular morphological changes and increased activation of the pro-apoptotic marker caspase-3 in Stx2/LPS mice. The findings that tubular dysfunction occurred in Stx2/LPS mice were also confirmed by an altered distribution of megalin in proximal tubules.

A number of studies have provided important information about the detrimental role of complement activation and the development of acute kidney injury [58,59,60,61]. In patients with evidence of acute kidney injury, C3 deposited in many of the injured tubules [58], whereas very high levels of complement activated fragments were found in plasma and urine of pediatric subjects [59]. Moreover, in a model of renal ischemia-reperfusion injury, the C3aR deficiency protected against renal functional impairment and tubular injury, in terms of decreased tubule thinning and dilatation, loss of brush borders and protein casts [60]. The hypothesis that C3a is a key mediator for acute tubular injury in HUS is based on our data that C3a inhibition reduced apoptosis and restored megalin distribution in the proximal tubular cells, demonstrating that the activation of the C3a/C3aR signaling is not just an accompanying epiphenomenon but contributes directly to the pathogenesis of renal tubular injury in Stx2/LPS mice.

To gain a better understanding of the mechanisms underlying complement-mediated renal injury in HUS, we focused on the impact that the activation of the C3a/C3aR axis had on the dysregulation of the mitochondrial functions. We found that in Stx2/LPS mice, the ultrastructure and distribution of mitochondria were altered in podocytes and, to a greater extent, in proximal tubular cells, with reduced mitochondrial matrix density and irregular cristae. Finding that VDAC and ATP5I were reduced in Stx2/LPS mice may suggest reduced mitochondrial mass and altered organelle permeability as well as the loss of cristae arrangement and energy production. Indeed, it is known that VDAC, the most abundant protein on the outer mitochondrial membrane, is the gatekeeper for mitochondria permeability and membrane potential, mediating the passages of metabolites, nucleotides and ions [37]. On the other hand, ATP5I is a subunit of ATPase that stabilizes F1F0 ATP synthase dimers at the crista rim, thus regulating ATP production in mitochondria through the oxidative phosphorylation, as well as the formation and stabilization of mitochondrial cristae [38]. In our setting, the hypothesis that C3a plays a deleterious role in triggering mitochondrial dysfunction rests on data showing that C3aR blockade preserved mitochondrial morphology, reducing the appearance of dysmorphic mitochondria with irregular cristae. The inhibition of C3a, by normalizing VDAC and ATP5I levels, also limited mitochondrial functional alterations, which are instrumental to cell apoptosis. On the basis of our data, however, we cannot rule out the possibility that cytosolic ATP production and extracellular ATP release may be modulated in Stx2/LPS mice, possibly through C3a, as other groups have described [62,63]. These findings in experimental HUS are consistent with our previous data, which showed the detrimental role of C3a on podocytes, through mitochondrial dysfunction, in mice with diabetic nephropathy [32]. 

The above findings are corroborated by in vitro evidence that C3a is a direct effector of mitochondrial damage as shown by the alterations of mitochondrial dynamics towards fission/fragmentation through the C3a/C3aR axis. In further support to this mechanism is the redistribution of mitochondria at the perinuclear level in response to C3a in vitro, as well as in vivo, that could be due to altered microtubulin network, known to regulate intracellular mitochondrial dislocation and function [41]. Consequent to mitochondrial structural alterations, C3a engagement also affected the mitochondrial antioxidant defense of both renal cell populations, pointing to C3a as a critical mediator for mitochondrial-dependent oxidative stress, known to be instrumental for the induction of pro-apoptotic pathways [64], in the kidney of Stx2/LPS mice. Finally, our data that Stx2 was able to enhance glomerular and tubular cell sensitivity to the adverse effects of C3a through the overexpression of its specific receptor are in strong agreement with the notion that Stx2 directly regulated complement activation, amplifying the detrimental effect of C3a/C3aR signaling at the renal level in HUS. 

Our data on the renal protective effects of treatment with C3aR antagonist in the experimental model of HUS induced by Stx2/LPS point to the clinical translational relevance of this pharmacological approach. Should the present findings be documented in patients with Stx-associated HUS, drugs inhibiting C3aR or intercepting C3 would represent a new therapeutic strategy for the clinical management of this disease.

## Figures and Tables

**Figure 1 cells-11-01755-f001:**
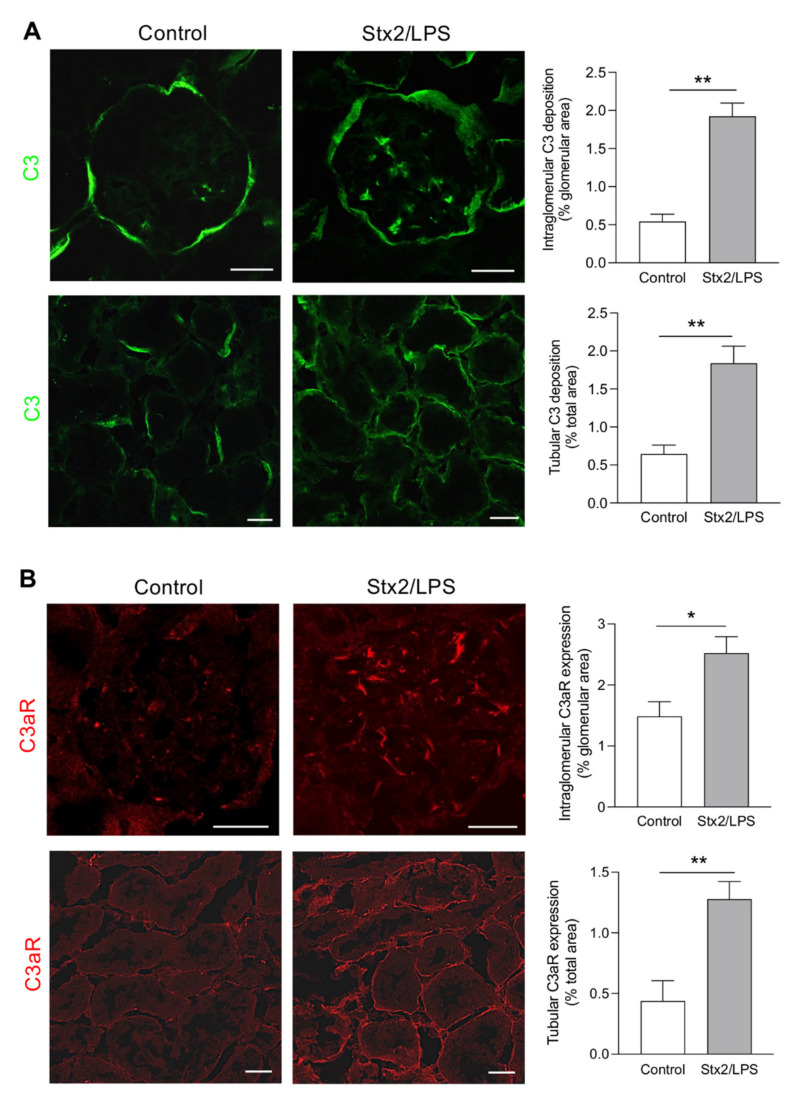
Glomerular and tubular C3 and C3aR staining in Stx2/LPS mice: (**A**) Representative images and quantification of intraglomerular and tubular C3 staining (green) in control and Stx2/LPS-injected mice at 48 h; (**B**) Representative images and quantification of intraglomerular and tubular C3aR expression (red) in control and Stx2/LPS-injected mice. Scale bars: 20 μm. Results are presented as mean ± SEM (control *n* = 4, Stx2/LPS *n* = 7), and unpaired Student’s *t* test was used. * *p* < 0.05, ** *p* < 0.01.

**Figure 2 cells-11-01755-f002:**
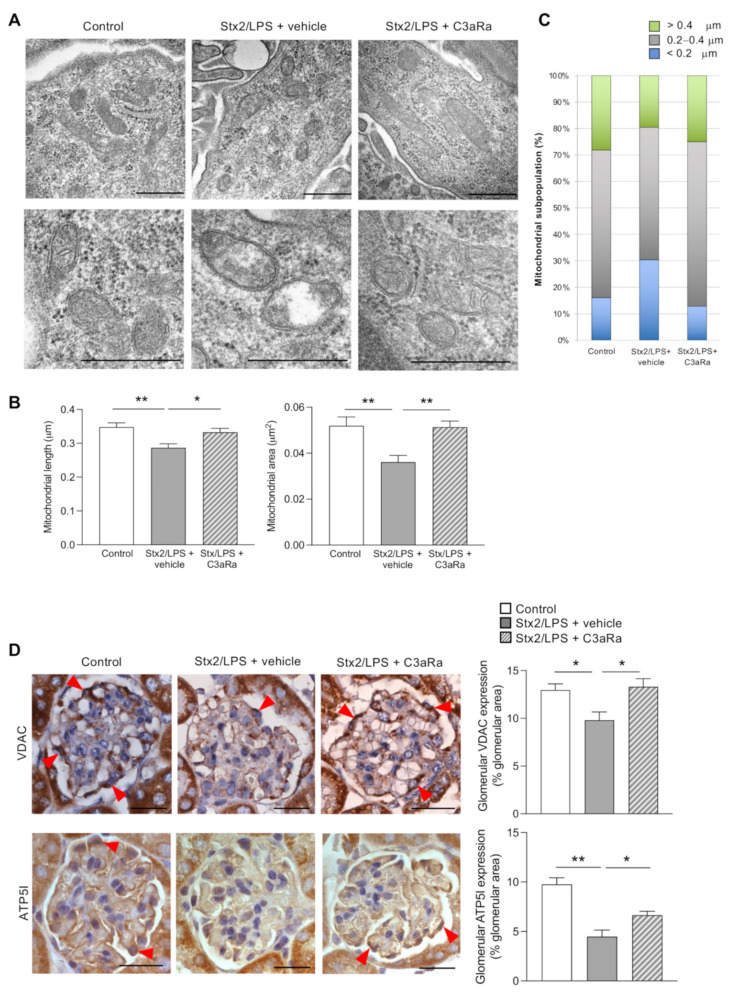
Treatment with the C3aR antagonist limits glomerular mitochondrial alterations in Stx2/LPS mice: (**A**) Representative transmission electron microscope images showing mitochondrial morphology in podocytes of control and Stx2/LPS mice, given vehicle or C3aR antagonist (C3aRa). Scale bars: 500 nm; (**B**) Quantification of mitochondrial length and area and (**C**) characterization of mitochondrial subpopulations in podocytes. Results are expressed as mean ± SEM (control: *n* = 516 mitochondria analyzed in *n* = 6 glomeruli of *n* = 2 mice; Stx2/LPS + vehicle: *n* = 411 mitochondria analyzed in *n* = 7 glomeruli of *n* = 3 mice; Stx2/LPS + C3aRa: *n* = 529 mitochondria analyzed in *n* = 7 glomeruli of *n* = 3 mice); (**D**) Representative images and quantification of glomerular VDAC (upper panels) and ATP5I (bottom panels) staining in controls and Stx2/LPS mice given vehicle or C3aRa. Podocytes are indicated by red arrowheads. Scale bars: 20 μm. Results are expressed as mean ± SEM (VDAC: control *n* = 4, Stx2/LPS + vehicle *n* = 5, Stx2/LPS + C3aRa *n* = 4; ATP5I: control *n* = 4, Stx2/LPS + vehicle *n* = 7, Stx2/LPS + C3aRa *n* = 7), and ANOVA with Tukey multiple-comparisons test was used. * *p* < 0.05, ** *p* < 0.01.

**Figure 3 cells-11-01755-f003:**
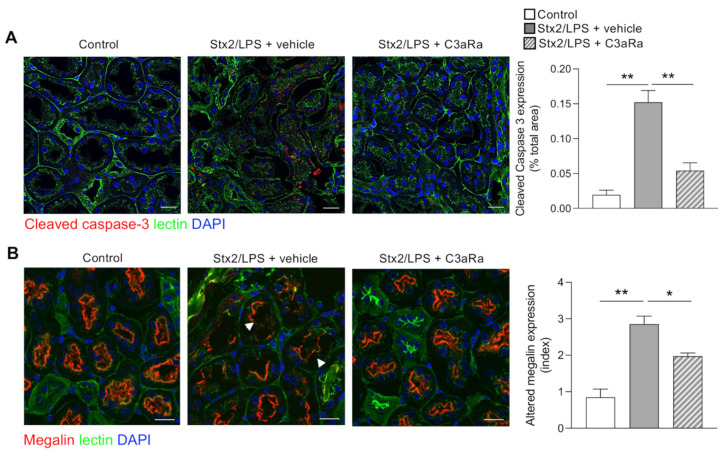
The C3aR blockade reduced proximal tubular cell apoptosis and restored megalin expression in Stx2/LPS mice. (**A**,**B**) Representative images and quantification of (**A**) cleaved caspase-3 (red) and (**B**) megalin (red) in control and Stx2/LPS mice given vehicle or C3aRa. Arrowheads indicate loss of megalin expression on the proximal tubular cell brush border. Cell membranes and nuclei were stained with FITC-WGA-lectin (green) and DAPI (blue), respectively. Scale bars: 20 μm. Results are expressed as mean ± SEM (*n* = 4 per each group), and ANOVA with Tukey multiple-comparisons test was used. * *p* < 0.05, ** *p* < 0.01.

**Figure 4 cells-11-01755-f004:**
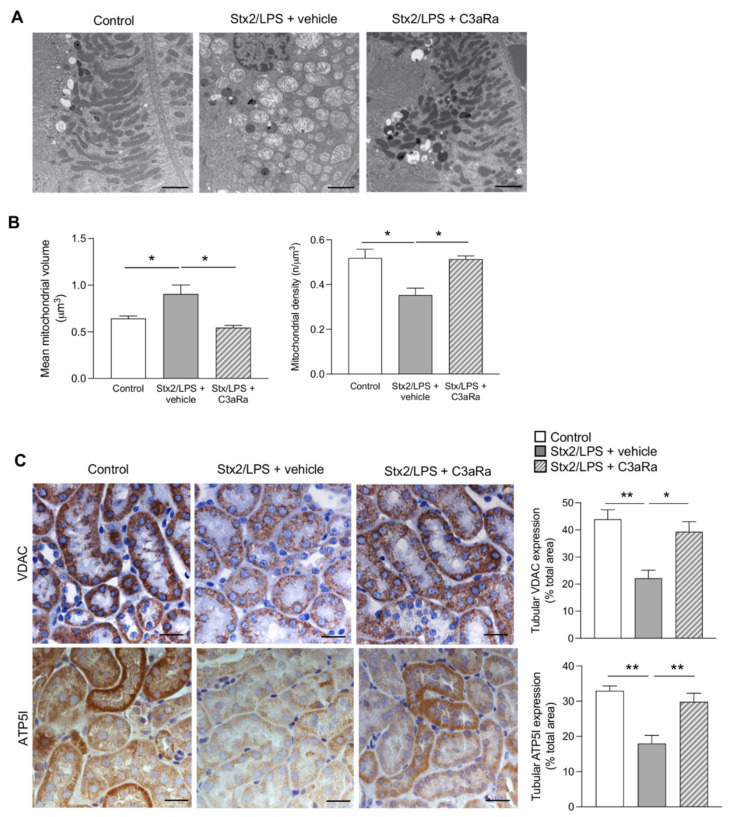
The C3aR blockade restores mitochondrial morphology and function in proximal tubular cells in Stx2/LPS mice. (**A**,**B**) Representative transmission electron microscope images and morphometric analysis showing mitochondrial morphology in proximal tubular cells of control and Stx2/LPS mice, given vehicle or C3aRa. Scale bars: 500 nm. Results are expressed as mean ± SEM (control: *n* = 6768 mitochondria analyzed in *n* = 3 mice; Stx2/LPS + vehicle: *n* = 4238 mitochondria analyzed in *n* = 3 mice; Stx2/LPS + C3aRa: *n* = 7100 mitochondria analyzed in *n* = 3 mice). (**C**) Representative images and quantification of tubular VDAC (upper panels) and ATP5I (bottom panels) stainings in control and Stx2/LPS mice with or without C3aRa. Scale bars: 20 μm. Results are expressed as mean ± SEM (VDAC: control *n* = 4, Stx2/LPS + vehicle *n* = 5, Stx2/LPS + C3aRa *n* = 4; ATP5I: control *n* = 4, Stx2/LPS + vehicle *n* = 7, Stx2/LPS + C3aRa *n* = 7). ANOVA with Tukey multiple-comparisons test was used. * *p* < 0.05, ** *p* < 0.01.

**Figure 5 cells-11-01755-f005:**
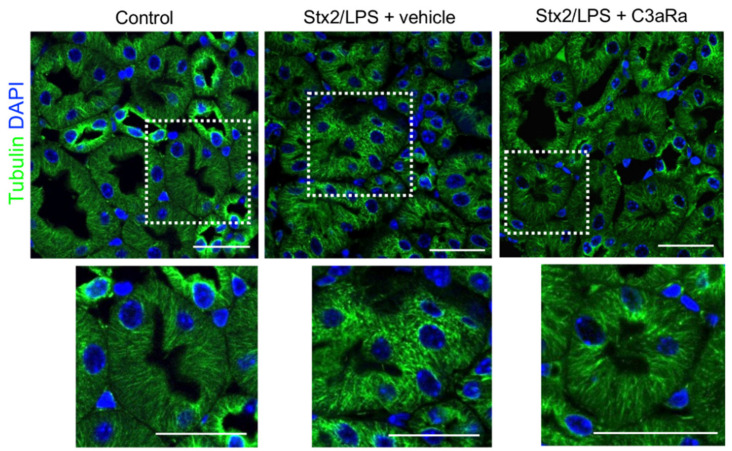
Treatment with the C3aR antagonist limits tubulin alterations in proximal tubules of Stx2/LPS mice. Representative images of tubulin expression (green) in control and Stx2/LPS mice receiving vehicle or C3aRa. Nuclei were stained with DAPI (blue). Scale bars: 20 μm.

**Figure 6 cells-11-01755-f006:**
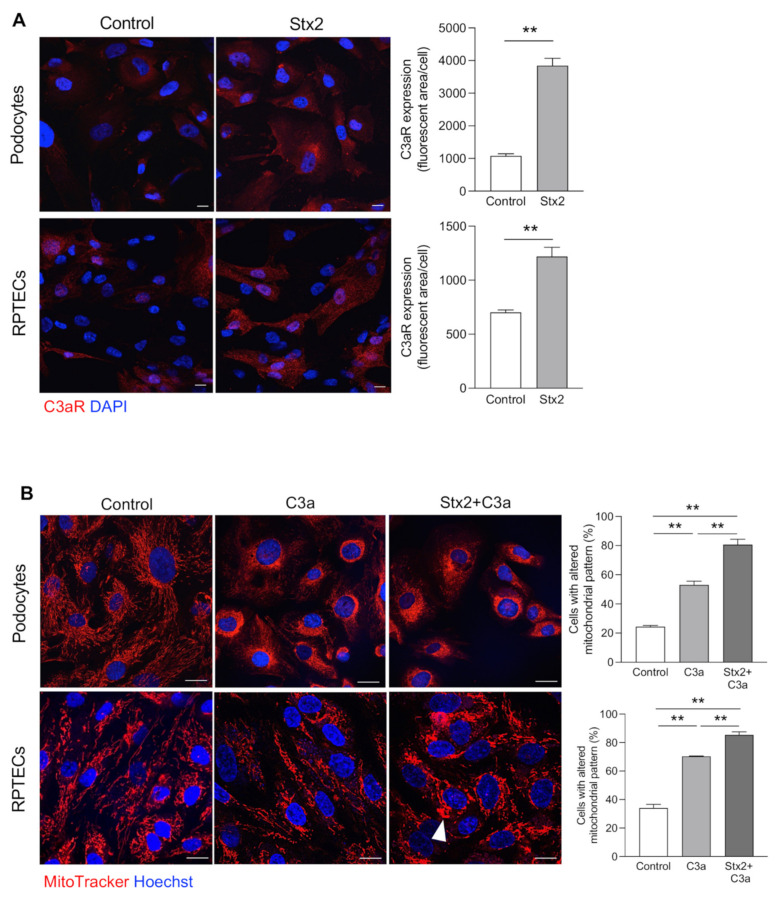
Stx2 increases podocyte and tubular cell sensitivity to C3a through the upregulation of C3aR expression in vitro. (**A**) Representative images and quantification of C3aR staining (red) in podocytes and RPTECs exposed to control medium or Stx2 (50 pM) for 15 h. Nuclei were counterstained with DAPI (blue). Scale bars: 20 μm. Results are presented as mean ± SEM (number of biological samples: *n* = 3 control and Stx2-treated podocytes; *n* = 4 control RPTECs and *n* = 6 Stx2-treated RPTECs), and unpaired Student’s *t* test was used. (**B**) Representative images of mitochondria labeled with MitoTracker in live podocytes and RPTECs incubated with control medium, C3a (1 μM, 6 h) alone, or with Stx2 (50 pM, 24 h) in the presence of C3a, added in the last 6 h. Nuclei were counterstained with Hoechst (blue). Arrowhead indicates mitochondrial swelling. The percentage of cells with an altered mitochondrial pattern, in terms of fragmentation and perinuclear redistribution, on total cells per field was quantified. Scale bars: 20 μm. Results are expressed as mean ± SEM (number of biological samples: *n* = 3 control, C3a and Stx2 + C3a-treated podocytes and RPTECs), and ANOVA with Tukey multiple-comparisons test was used. ** *p* < 0.01.

**Figure 7 cells-11-01755-f007:**
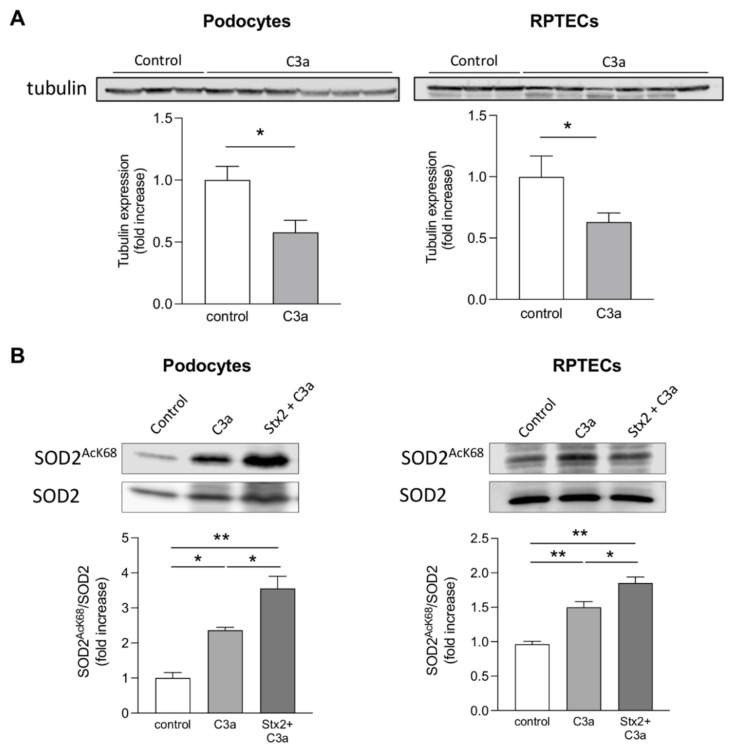
C3a decreases tubulin expression and reduces mitochondrial antioxidant defense in cultured podocytes and RPTECs. Representative western blot and densitometric analysis of (**A**) tubulin and (**B**) acetylated SOD2 (SOD2^AcK68^) and total SOD2 protein expression in protein extracts obtained from podocytes and RPTECs incubated with control medium, C3a (1 μM, 6h) alone, or with Stx2 (50 pM, 24 h) in the presence of C3a, added in the last 6 h. SOD2 acetylation was expressed as the ratio between SOD2^AcK68^ and total SOD2. Results are presented as mean ± SEM (number of biological samples for tubulin: *n* = 3 control and *n* = 6 C3a-treated podocytes; *n* = 6 control and *n* = 12 C3a-treated RPTECs), (number of biological samples for SOD2: *n* = 3 control, C3a and Stx2+C3a-treated podocytes and RPTECs), and unpaired Student’s *t* test or ANOVA with Tukey multiple-comparisons test were used. * *p* < 0.05, ** *p* < 0.01.

## Data Availability

The original contributions presented in the study are included in the article/Appendix A. Further inquiries can be directed to the corresponding author.

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
