# Peer review of "Shiga Toxin 2 Triggers C3a-Dependent Glomerular and Tubular Injury through Mitochondrial Dysfunction in Hemolytic Uremic Syndrome"

_cells, 2022, doi:10.3390/cells11111755_

Round 1
Reviewer 1 Report
The paper by Buelli et al studied the role of Stx2 and C3a in a model of hemolytic uremic
syndrome and show that Stx-mediated C3a signaling is associated with podocyte and tubular
cell damage by the induction of mitochondrial dysfunction. These effects were reduced by a
C3a receptor antagonist. The data are potentially interesting but some more experiments and
explanations will be required as detailed below.
Major comments
1) How can the in vivo effects of LPS contra Stx2 be assessed when mice were only
treated with both and not with Stx2 or LPS alone? Particularly if the effects are
mostly due to LPS then this is not really a model of HUS, therefore the results from
LPS alone should also be shown.
2) Line 82: the doses of Shiga toxin 2 (120 ng/mouse) and LPS (60 μg/mouse) given to
each mouse are very high compared to other studies. Why did the authors choose
these doses? What signs of disease did they mice have and did any mice die before 48
hours?
3) Line 197: immunofluorescence for C3aR was carried out without an irrelevant
antibody, only omission of the primary antibody was used as a control. This is
insufficient as incubating the cells with Stx will induce injury which could lead to
unspecific antibody deposition.
4) Line 294: the quantification of mitochondrial changes in podocytes in Figure 2B is
unclear as the figure legend mentions the number of glomeruli, not the number of
mitochondria that were quantified. Lines 148-9 mention that at least 400-500
mitochondria were determined in podocytes of 6-7 glomeruli for 2-3 mice/group. This
is a rough estimation, please give the exact numbers in the figure legend. The same
relates to Figure 3B.
5) Data related to ATP5I are interesting. The interpretation that Shiga toxin reduces ATP
production is however not completely in line with previous publications which have
shown a toxin-mediated increase in ATP both in vitro and in vivo (PMID 31591425).
It seems that plasma ATP in mice is increased after Stx2 injection although this may
be due to the release of preformed cytosolic ATP. Also, C3aR activation induces the
release of cytosolic ATP after LPS stimulation (PMID 23878142). It has been
suggested that the release of ATP affects glycolysis (32840864) which could have a
secondary effect on mitochondria. The authors should therefore rephrase their
interpretation in the results and discussion.
6) It seems that mitochondria possess the C5aR, not the C3aR, how do the authors
propose C3a affects mitochondria?
7) Supplementary Figure S3 panels B and C should be included in the body of the paper.
The arrows in panel B were not explained.
8) In Figure 3 the authors studied mitochondrial volume, which was not assayed in
podocytes, why?
9) Experiments in Figure 5 and Figure 6 should include the C3aR antagonist to
demonstrate specificity of the effect of C3a on mitochondria. Also, the effect of Stx2
on SOD2 (Fig 6) should be shown.
Minor comments:
1) Stained samples from mice were quantified in 5, 10, 15 or 20 fields. Why were
different quantification fields chosen?
2) Line 79: what is the impact of breeding in a pathogen-free environment on this mouse
model?
3) Line 80: please describe the LPS content of the purified Shiga toxin 2.
4) Shiga toxin are two words this refers to the title and the abstract.
5) In the abstract Escherichia coli should be spelled with a little c in coli.
6) Line 59 and line 74 “HUS mice” or “mice with HUS” please change this to “in an
experimental model of HUS in mice” or “a murine model of HUS” as mice do not
actually develop HUS.
7) Line 86, The authors state that they gave the C3aR antagonist twice a day and then
mention that it was given at 1 hour before Stx/LPS and 4, 8, 23, hours after, those
times are 4 times in 24 hours. Please correct.
8) Line 142: Is Perico, Nat Commun 2017 a reference?
9) Supplementary Figure S2: The Y axis should be mg/dL, please correct. Also explain
that in C3aRa that last “a” means antagonist.
10) Supplementary Figure 3a: the figure legend does not explain what the figure shows,
the changes in tubules need to be quantified, the changes are not described and the
asterix is unclear.
11) Please check the English grammar throughout, for example line 273, the word “was”
(what was observed) is missing.
12) Line 368, what does n=4 and 6 mean?
10) Lines 186 and 187 mention a Stx concentration of 50 pM for in vitro experiments.
Line 359 mentions only 100 pM. Please correct. How was this concentration chosen
and what effect does it have on cell viability at 24 h?
11) The results in Figure 4 are very nice, though the statement that the C3aR antagonist
reorganized the tubulin distribution is not appropriate, it abrogated the disruptive
effect that Stx/LPS had on tubulin.
13) The sentence on line 478-9 requires a reference.
14) Line 23 of the abstract, upregulation would often refer to mRNA but data in Figure 5
show increase at the protein level, this could be rephrased.
15) Line 146: was mitochondrial size estimated based on electron microscopy?
Reviewer 2 Report
The paper by Buelli et. al. describes that treatment of mice with Shiga-toxin 2 (Stx2) triggers C3a-dependent glomerular and tubular damage in a HUS model. The authors show that inyección of mice with Stx2/LPS induced glomerular and podocyte C3 deposits and C3aR activation associated with cell damage. This C3a activation affected mitochondrial ultrastructure wellness and function and blockade of C3aR prevented these alterations as well as damage. The authors propose that Stx2 triggers renal injury by inducing mitochondria disfunction through the C3a/C3aR axis.
The manuscript is straightforward, the methods and results are clear and robust and the hypothesis and conclusions supported by the background and data presented. I have only one concern. The mouse HUS model is triggered by injecting Stx2/LPS but the control group is treated with saline. Why not use saline with LPS as a control? If C3a/C3aR accumulation is Stx2 dependent than saline/LPS should not have any effect. With this setting it is hard to separate the effect of the LPS from the one triggered by Stx2. Maybe the authors could show that some of these effects are not observed when LPS is injected, particularly the C3a accumulation as it is known that LPS induces complement activation. Or maybe this can be found in the literature and I am not aware of it. In the in vitro experiments only Stx2 was used to show C3a accumulation and mitochondrial abnormalities.
Reviewer 3 Report
The paper entitled “Shigatoxin 2 triggers C3a-dependent glomerular and tubular injury through mitochondrial dysfunction in Hemolytic Uremic Syndrome” by Buelli and colleagues is a novel and interesting work that supports the involvement of the activation of complement system in HUS pathophysiology. The work focuses on the effects of C3a complement component on podocyte and tubular damages in a mice model of HUS and on cultured human podocytes and tubular cells. The authors study the mechanisms by which C3a triggered by Stx2/LPS may induce glomerular damages. In this sense they conclude that activation of C3a/C3aR signaling in podocytes and tubular cells induces mitochondrial dysfunction and cell injury. The authors suggest that a C3aR antagonist or a C3 intercepting agent may represent a novel therapeutic strategy for the clinical managment of Stx/associated HUS.
I consider that the results contribute to the knowledge in the pathophysiology of HUS. I had a few comments about the manuscript.
Major comments
- The authors use an HUS mice model obtained by the coinjection of Stx2 and LPS. LPS may also activate complement. How authors can confirm that Stx2 is mainly responsible for triggering complement activation in this model? Stx2 only injected mice will be important to understand the contribution of Stx2 or LPS to the complement activation. In this sense, the discussion of the contribution of LPS in complement activation in the introduction and discussion sections may enrich the work.
- Is the HUS model a lethal or sublethal model? Stx2 dose variation may vary the intensity of complement activation?
Minor comments
- Figure 2C. Length of mytochondria is expressed in nm. I think that probably there is a mistake and the value may be in µm. Also see in lines 271-273.
- In Figure 2C the lower range of mitochondria length goes from 0 to 0.2. Since 0 is not a real possible value I suggest to name them as: “>4; 0.2-0.4 and < 0.2”.
- Figure 6. Bold in the title of the legend in missing
Round 2
Reviewer 2 Report
The authors have answered my concerns.